# Align then Fusion: Generalized Large-scale Multi-view Clustering with Anchor Matching Correspondences

**Siwei Wang[1], Xinwang Liu[1,*], Suyuan Liu[1], Jiaqi Jin[1], Wenxuan Tu[1], Xinzhong Zhu[2], En Zhu[1]**
[1]School of Computer, National University of Defense Technology, Changsha, China
[2]Zhejiang Normal University
`{wangsiwei13, xinwangliu, suyuanliu, jinjiaqi, twx, enzhu}@nudt.edu.cn`

## Abstract

Multi-view anchor graph clustering selects representative anchors to avoid full pair-wise similarities and therefore reduce the complexity of graph methods. Although widely applied in large-scale applications, existing approaches do not pay sufficient attention to establishing correct correspondences between the anchor sets across views. To be specific, anchor graphs obtained from different views are not aligned column-wisely. Such an **A**nchor-**U**naligned **P**roblem (AUP) would cause inaccurate graph fusion and degrade the clustering performance. Under multi-view scenarios, generating correct correspondences could be extremely difficult since anchors are not consistent in feature dimensions. To solve this challenging issue, we propose the first study of the generalized and flexible anchor graph fusion framework termed **F**ast **M**ulti-**V**iew **A**nchor-**C**orrespondence **C**lustering (FMVACC). Specifically, we show how to find anchor correspondence with both feature and structure information, after which anchor graph fusion is performed column-wisely. Moreover, we theoretically show the connection between FMVACC and existing multi-view late fusion [18] and partial view-aligned clustering [7], which further demonstrates our generality. Extensive experiments on seven benchmark datasets demonstrate the effectiveness and efficiency of our proposed method. Moreover, the proposed alignment module also shows significant performance improvement applying to existing multi-view anchor graph competitors indicating the importance of anchor alignment. Our code is available at `https://github.com/wangsiwei2010/NeurIPS22-FMVACC`.

## 1 Introduction

As an effective unsupervised multi-view learning technology, multi-view graph clustering (MVGC) comprehensively utilizes multiple pair-wise instance similarities into optimal flexible graph structures [26, 23, 5, 41]. In general, MVGC firstly constructs graph structures for every single view and then refines individual graphs with ideal fused graph [44, 40, 43, 45]. For example, Zhang et al. explore latent representation from multiple views with linear models and nonlinear networks in [44]. [40] optimizes the consensus graph by minimizing disagreement of individual graphs and then reach an agreement with rank constraint. [45] further proposes to utilize partition level information rather than graph structure agreement and achieve promising performance.

Although massive MVGC approaches have been proposed, one major issue is scalability in real large-scale applications [26, 23, 5]. For existing MVGC approaches, the complexity is quadratic or cubic respecting to instance number $n$, which is unbearable for big data [9, 36]. As an effective way for large-scale problem, anchor graph strategy firstly selects $m$ individual anchors to represent full

---

*Corresponding author

36th Conference on Neural Information Processing Systems (NeurIPS 2022).

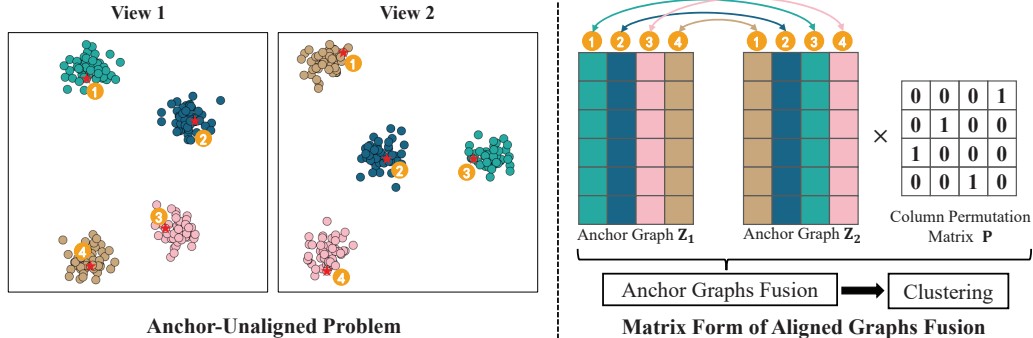

Figure 1: An example of the Anchor-Unaligned Problem (AUP) for two views. (Left) AUP of view 1 and view 2: wrong correspondences between anchor sets; (Right) Columns of anchor graphs $\mathbf{Z}_1$ and $\mathbf{Z}_2$ are not orderly arranged, which often leads to inaccurate graph fusion in clustering tasks. Therefore, it is necessary to find correct correspondences before fusion and then align them.

instances among views [14, 31, 11, 13, 27, 34, 32]. Generally, the quality of anchors is of central importance for performance and the sampling and $k$-means two strategies are commonly-used. For example, Kang et al. operate $k$-means on each view separately and fuse into an optimal graph [11]. [34] proposes to jointly optimize anchors and the respective graph among multiple views. The space and time complexity have been reduced into $\mathcal{O}(vnm)$ and $\mathcal{O}(nm^2)$ linear to instance numbers, which proves to be efficient for large-scale tasks [17].

Despite being widely applied in large-scale applications, one vital factor of successful multi-view anchor graph clustering is to build correct correspondences for anchor sets. The selected anchor sets generated by different views may mismatch without guarantees of correspondences, since $k$-means or sampling is performed on each view separately. Taking two anchor graphs $\mathbf{Z}_1 \in \mathbb{R}^{n \times m}$ and $\mathbf{Z}_2 \in \mathbb{R}^{n \times m}$ as showcase in Fig. 1, the anchor graphs are not aligned column-wisely, which we refer as **A**nchor-**U**naligned **P**roblem (AUP). Such an unaligned issue would cause inappropriate graph fusion and degrade the clustering performance in return. One pioneer work, SFMC [13] provides an intuitive way to select samples with same indexes to implicitly avoid wrong correspondences. However, this way could destroy the flexibility of anchors. To the best of our knowledge, no generalized framework for flexible multi-view anchor correspondences has been proposed so far since it is difficult to directly compute anchor distances with inconsistent dimensions for different views.

To this end, we propose a generalized and flexible multi-view anchor graph fusion framework termed **F**ast **M**ulti-**V**iew **A**nchor-**C**orrespondence **C**lustering (FMVACC) in this paper. Unlike the existing rigid style of fixed indexes, FMVACC flexibly selects anchors with more discriminate inner-view structures. Then, we elegantly solve the unmeasurable multi-dimensional anchor issue by relational representations. Based on this, FMVACC captures both feature and structure information to help establish accurate anchor correspondences. After that, the anchor graph fusion is column-wisely performed with cross-view consistency. Further, we show that the late fusion [18] and the PVC [7] are our special variants. Extensive experiments clearly demonstrate the effectiveness of our method. Moreover, our proposed alignment module also shows significant clustering performance improvement applying to existing multi-view anchor graph competitors, which indicating the importance of anchor alignment. We summarize the contributions of this work as follows,

- We study a new paradigm for large-scale multi-view anchor graph clustering, termed as **A**nchor-**U**naligned **P**roblem (AUP). The selected anchor sets in multi-view data are not aligned, which may lead to inaccurate graph fusion and degrade the clustering performance.

- We propose a flexible anchor graph fusion framework termed FMVACC to tackle the AUP problem. After generating flexible anchor candidates, an anchor alignment module is proposed to solve AUP with both feature and structure information. To the best of our knowledge, it is the first study of the flexible anchor correspondence fusion framework.

- Extensive experiments demonstrate the effectiveness and efficiency of our proposed framework. Moreover, the proposed alignment module also shows significant performance improvement on existing approaches which indicates the importance of anchor alignment.

## 2 Related Work

### 2.1 Multi-view Graph Clustering

As a representative unsupervised multi-view learning method [6, 37, 16, 29, 47], multi-view graph clustering (MVGC) represents single-view structures with graphs and then operates graph fusion on individual graphs [15, 28, 24, 3, 42]. Many graph approaches have been proposed with the differences of these two parts. For constructing stage, self-expressive representation and local similarity graph are proved to be effective on finding pairwise relationships [4, 45, 2, 44, 43, 40]. Moreover, by imposing various regularization terms on the optimal graph, graph fusion stage reach consensus agreement on similarity and partition information [22, 10, 8, 20]. Zhan et al. optimize the consensus graph by minimizing disagreement of individual graphs and then optimizes with low-rank constraint [40]. Notice that the AUP problem does not happen under full pair-wise graph settings since the anchors are instances themselves and therefore naturally aligned.

### 2.2 Multi-view Anchor Graph Clustering

Although existing multi-view graph clustering methods have been efficient in improving the performance, the high computational complexity prevents their application to large-scale datasets. The space and time complexity of majority graph approaches are $\mathcal{O}(vn^2)$ and $\mathcal{O}(n^3)$ quadratic and cubic to instance numbers [25, 14]. To tackle the above issue, multi-view anchor graph clustering has been widely studied to effectively reduce the time and space consumption by constructing anchor graphs with selected $m$ anchors instead of the entire instances [30].

Specifically, the first step of multi-view anchor graph clustering is to construct individual anchor graphs on each view by solving the following problem,

$$\min_{\mathbf{Z}_i} \|\mathbf{X}_i - \mathbf{Z}_i\mathbf{A}_i\|_\mathbf{F}^2 + \beta\|\mathbf{Z}_i\|_\mathbf{F}^2, \text{ s.t. } \mathbf{Z}_i \geq 0, \mathbf{Z}_i\mathbf{1}_m = \mathbf{1}_n, \tag{1}$$

where $\mathbf{A}_i \in \mathbb{R}^{m \times d_i}$ is the anchor matrix for the $i$-th view. $\mathbf{Z}_i \in \mathbb{R}^{n \times m}$ is the anchor graph for the $i$-th view and $\mathbf{Z}_i\mathbf{1}_m = \mathbf{1}_n$ ensures that the similarity sum of each sample to all the anchors equals 1. Noticed that $\mathbf{A}_i$ is fixed before the construction process, so the quality of anchors plays an essential part in the success of multi-view anchor graph clustering. Li *et al.* [14] and Kang *et al.* [11] propose to get the anchor set by performing $k$-means on the original data of individual views. Random sampling is also a common strategy for selecting anchor points [21]. Moreover, Li *et al.* [13] provide a directly alternate sampling method to determine the anchors based on scores.

After constructing the view-specific anchor graphs $\{\mathbf{Z}_i\}_{i=1}^v$, the second step is to get the fused anchor graph $\mathbf{G} \in \mathbb{R}^{n \times m}$ as follows,

$$\min_{\boldsymbol{\alpha}, \mathbf{G}} \left\|\sum_{i=1}^v \alpha_i\mathbf{Z}_i - \mathbf{G}\right\|_\mathbf{F}^2, \text{ s.t. } \mathbf{G}\mathbf{1}_m = \mathbf{1}_n, \boldsymbol{\alpha}^\top\mathbf{1}_v = 1, \boldsymbol{\alpha} \geq \mathbf{0}, \tag{2}$$

where $\boldsymbol{\alpha} \in \mathbb{R}^v$ is the weight coefficient which measures the impact of each view and $\boldsymbol{\alpha}^\top\mathbf{1}_v = 1$ guarantees that $\mathbf{G}\mathbf{1}_m = \mathbf{1}_n$. The weight coefficient $\boldsymbol{\alpha}$ and fused graph $\mathbf{G}$ are alternatively optimized in a unified framework. However, solving Eq. (2) will lead to a trivial solution. To avoid the above problem, Kang *et al.* [11] directly assign the same weight to all views and Li *et al.* [13] further solve it by imposing rank constraint on the fused anchor graph. After obtaining the fusion graph $\mathbf{G}$, we perform $k$-means on its left singular vector to get the final clustering result. The space and time complexity of multi-view anchor graph clustering have been reduced into $\mathcal{O}(vnm)$ and $\mathcal{O}(nm^2)$, which can be applied to large-scale tasks effectively.

## 3 Anchor-Unaligned Clustering for Large-scale Multi-view Data

In this section, we propose a generalized and flexible anchor graph fusion framework, termed fast multi-view anchor-correspondence clustering (FMVACC), which could elegantly solve the anchor-unmeasurable issue for different views and therefore perform anchor graph fusion correctly. We firstly show how FMVACC captures the feature and structure information for multi-view anchor set matching. Then we present our FMVACC framework with theoretical analysis and show the connection between late fusion strategy and partial view-aligned clustering.

**Algorithm 1** Obtain single-view anchors and anchor graphs

---

**Input:** Multi-view dataset $\{\mathbf{X}_i\}_{i=1}^v$ and anchor number $m$.
1: Initialize $m$ anchor points in the $i$-th view.
2: **repeat**
3:     Update $\{\mathbf{Z}_i\}_{i=1}^v$ by solving Eq. (4);
4:     Update $\{\mathbf{A}_i\}_{i=1}^v$ by solving Eq. (5);
5: **until** converged.
**Output:** $\{\mathbf{A}_i\}_{i=1}^v$ and $\{\mathbf{Z}_i\}_{i=1}^v$ as the obtained anchors and respective anchor graphs.

---

### 3.1 Flexible Anchor Generation

Given multi-view data matrices $\{\mathbf{X}_i\}_{i=1}^v$ with $n$ samples, $v$ views and $d_i$ dimension for the $i$-th view. By selecting $m$ anchors in each view and following anchor graph framework described in Eq. (1) in Section 2.2, we define the obtained anchor set and the respective individual anchor graph for the $i$-th view are $\mathbf{A}_i \in \mathbb{R}^{m \times d_i}$ and $\mathbf{Z}_i \in \mathbb{R}^{n \times m}$.

Firstly, it is widely accepted that the quality of anchors plays an essential part in the success of multi-view anchor graph clustering, and adaptive optimizing anchors is proven to be effective in clustering contrary to existing fixed strategy. In our method, different from no constraint of $\{\mathbf{A}_i\}_{i=1}^v$ in Eq. (1), the anchors are further imposed to be orthogonal that $\mathbf{A}_i\mathbf{A}_i^\top = \mathbf{I}_m$ in each view to make the learned anchors more discriminative and diverse.

$$\min_{\mathbf{Z}_i, \mathbf{A}_i} \|\mathbf{X}_i - \mathbf{Z}_i\mathbf{A}_i\|_{\mathbf{F}}^2 + \beta\|\mathbf{Z}_i\|_{\mathbf{F}}^2, \text{ s.t. } \mathbf{A}_i\mathbf{A}_i^\top = \mathbf{I}_m, \mathbf{Z}_i \geq 0, \mathbf{Z}_i\mathbf{1}_m = \mathbf{1}_n. \tag{3}$$

The optimization of Eq. (3) can be solved by a two-step alternative approach as follows,

**Updating $\mathbf{Z}_i$:** Denoting $z_{j,l}^{(i)}$ as the $l$-th element in $j$-th row, Eq. (3) can be solved by row as the following form,

$$\min_{\mathbf{z}_{j,:}^{(i)}} \left\|\mathbf{z}_{j,:}^{(i)} - \mathbf{y}_{j,:}^{(i)}\right\|_{\mathbf{F}}^2, \text{ s.t. } \mathbf{z}_{j,:}^{(i)} \geq \mathbf{0}, \mathbf{z}_{j,:}^{(i)}\mathbf{1}_m = 1, \tag{4}$$

where $\mathbf{y}_j^{(i)} = (\mathbf{X}_i\mathbf{A}_i^\top)_{j,:}/(1+\beta)$. Eq. (4) is a projection capped simplex problem defined in [35], which can be solved efficiently at global minimum with $\mathcal{O}(nmd_i)$.

**Updating $\mathbf{A}_i$:** By denoting $\mathbf{B} = \mathbf{Z}_i^\top\mathbf{X}_i$, updating anchor matrices $\{\mathbf{A}_i\}_{i=1}^v$ is equivalent as follows,

$$\max_{\mathbf{A}_i} \text{Tr}(\mathbf{A}_i^\top\mathbf{B}), \text{ s.t. } \mathbf{A}_i\mathbf{A}_i^\top = \mathbf{I}_m. \tag{5}$$

The optimum of $\mathbf{A}_i$ can be analytically obtained with rank $m$ truncated SVD by [33]. The complexity to update each $\mathbf{A}_i$ is $\mathcal{O}(nm^2 + m^2d_i)$. If $m > d_i$, we simply remove the orthogonal constraints into unconstrained optimization problems.

After optimization, we can obtain flexibly more representative single view anchors $\{\mathbf{A}_i\}_{i=1}^v$ and anchor graphs. However, the anchor sets $\{\mathbf{A}_i\}_{i=1}^v$ for individual views may **mismatch** with other views' counterparts, and therefore the formed anchor graphs are not aligned column-wisely. To achieve correct anchor graph clustering, we first need to establish the matching correspondences of the pair of anchor sets by applying graph matching approaches, which mathematically means to seek the matching correspondence or the permutation matrix $\mathbf{P} \in \{0,1\}^{m \times m}$ shown in Fig. 1. After that, anchor graph fusion is column-wisely performed the same as the conventional pipelines.

### 3.2 Generalized Multi-view Anchor Matching Framework

The main difficulty of solving the AUP problem is **unmeasurable anchor sets under multi-dimensional metric space**. The obtained anchors in multiple views are naturally presented with different dimensions, making it difficult to directly compute their distances and unmeasurable in various metric spaces. Therefore, a significant problem arises: **how to flexibly find correspondences for multiple anchor sets under different data metric spaces?**

One intuitive solution to implicitly avoid correspondence issues, proposed in SFMC [13], is to sample representative instances with the same indexes among multiple views. However, this sampling style

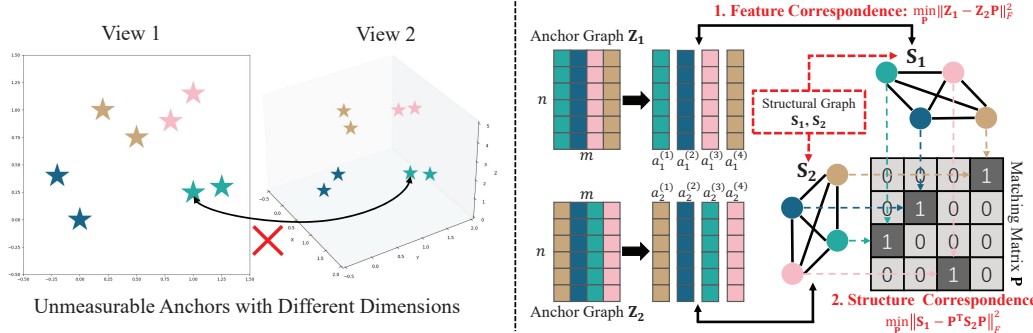

Figure 2: Overview of the proposed FMVACC consisting of two parts: feature and structure correspondence. (Left)Anchor sets are represented by different dimensions, which makes them **unmeasurable** in various metric spaces; (Right)In FMVACC, each column of the graph is taken as a new $n$-dimensional feature of each anchor, and the optimal matching matrix $\mathbf{P}$ is obtained by minimizing the both the feature and structure correspondences.

could destroy the flexibility and effectiveness of anchors and ignore the local individual view structure for clustering. It has been shown that not sampling but learning anchors in individual views has achieved more satisfactory performance, demonstrating the necessity of introducing flexibility into anchor frameworks. Another solution is to project multi-dimensional data into a commonly-shared space and then explicitly optimize consensus anchors in latent space [34]. However, the projection operator may lead to information loss and extra dimension hyperparameter selection. Flexible anchor sets naturally induce the unaligned matching problem, which is often ignored in the existing literature. To the best of our knowledge, no generalized framework of flexible anchor correspondences for multi-view clustering has not been formally proposed so far. Clearly, the clustering performance will be improved by establishing the correct correspondences of anchors during the fusion stage.

The first step is how to make the obtained multiple anchor sets $\{\mathbf{A}_i\}_{i=1}^{v}$ **measurable**. Going back to Eq. (3), the respective single view anchor graphs $\{\mathbf{Z}_i\}_{i=1}^{v}$ represent the similarities between the instances and the anchors. Elegantly, each column of the anchor graphs $\{\mathbf{Z}_i\}_{i=1}^{v} \subseteq \mathbb{R}^{n \times m}$ is taken as a new $n$-dimensional feature representation of each anchor, which makes the anchors between different views measurable. As shown in Fig. 2, we transform the anchor alignment problem into a graph matching problem of $m$ nodes with consensus $n$ embedding features. Under such transformation, the multiple anchor sets are measurable in $\mathbb{R}^n$ space and therefore can calculate their similarities to find set correspondences. Followed by the traditional graph matching problem [38], our proposed FMVACC retains both first-order (feature) and second-order (structure) information for better matching in the following subsections.

**Feature Correspondence:** For the first-order feature correspondence, we consider the principle that the correspondence probability of anchor points $a_1^{(i)}$ and $a_2^{(j)}$ should be high if their features are more similar than other pairs. $a_i^{(j)}$ is denoted as the $j$-th anchor in $i$-th view. Therefore, the feature correspondence can be fulfilled by minimizing the optimization goal as follows,

$$\min_{\mathbf{P}} \|\mathbf{Z}_1 - \mathbf{Z}_2\mathbf{P}\|_{\mathbf{F}}^2 \iff \max_{\mathbf{P}} \mathrm{Tr}(\mathbf{Z}_1^\top \mathbf{Z}_2 \mathbf{P}) = \max_{\mathbf{P}} \sum_{i=1}^{m} \sum_{j=1}^{m} \mathbf{K}_{ij} \mathbf{P}_{ji} = \max_{p} p^\top k,$$

$$\text{s.t. } \mathbf{P}\mathbf{1} = \mathbf{1}, \mathbf{P}^\top\mathbf{1} = \mathbf{1}, \mathbf{P} \in \{0,1\}^{m \times m}, \tag{6}$$

where $\mathbf{K} = \mathbf{Z}_1^\top \mathbf{Z}_2$, $p = \mathrm{vec}(\mathbf{P}^\top) \in \{0,1\}^{m^2}$, $k = \mathrm{vec}(\mathbf{K}) \in \mathbb{R}^{m^2}$ to denote the vectorization of matrices respectively. $\mathbf{K}_{ij}$ means the similarity between anchor pair $a_1^{(i)}$ and $a_2^{(j)}$ which can be as the opposite of the pair distances. It is easy to prove that by minimizing feature correspondence in Eq. (6), higher feature similarity value of $\mathbf{K}_{ij}$ will lead to bigger value of the assignment value $\mathbf{P}_{ji}$.

**Remark 1.** *Minimizing feature correspondence in Eq. (6) can be regarded as seeking the optimal transport plan for anchor sets $\mathbf{A}_1$ and $\mathbf{A}_2$. Define the pair distance $\mathbf{D}_{ij} = \mathrm{C} - \mathbf{K}_{ij}$, where $\mathrm{C}$ is a large enough number to ensure $\mathbf{D}_{ij} \geq 0$. Then $\max_{\mathbf{P}} \sum_{i=1}^{m} \sum_{j=1}^{m} \mathbf{K}_{ij} \mathbf{P}_{ji} = \max_{\mathbf{P}} m\mathrm{C} - \sum_{i=1}^{m} \sum_{j=1}^{m} \mathbf{D}_{ij} \mathbf{P}_{ji} \iff \min_{\mathbf{P}} \sum_{i=1}^{m} \sum_{j=1}^{m} \mathbf{D}_{ij} \mathbf{P}_{ji}$, where $\mathbf{P}$ indicates the optimal transport plan.*

---

**Algorithm 2** Fast Multi-View Anchor-Correspondence Clustering (FMVACC)

---

**Input:** Multi-view dataset $\{\mathbf{X}_i\}_{i=1}^v$, cluster number $k$ and anchor number $m$.
1: Obtain anchor sets $\{\mathbf{A}_i\}_{i=1}^v$ and respective anchor graphs $\{\mathbf{Z}_i\}_{i=1}^v$.
2: **for** $i = 2 \to v$ **do**
3:     Calculate $\mathbf{K} = \mathbf{Z}_1^\top \mathbf{Z}_i$ and $\mathbf{S}_1 = \mathbf{Z}_1^\top \mathbf{Z}_1, \mathbf{S}_i = \mathbf{Z}_i^\top \mathbf{Z}_i$ for Eq. (9);
4:     Obtaining permutation matrix $\mathbf{P}_i$ by solving Eq. (9)
5: **end for**
**Output:** Fused aligned graph $\mathbf{Z}_{Aligned}$ and permutation matrices $\{\mathbf{P}_i\}_{i=2}^v$.

---

**Structure Correspondence:** With the mentioned first-order feature correspondence part, we further consider the second-order graph structure regularization. To find better matching, the inner structures of anchor sets should also be comparable after matching, which indicates the matching between pair-wise edges. The second-order structure correspondence can be achieved by minimizing the following function,

$$\min_{\mathbf{P}} \left\| \mathbf{S}_1 - \mathbf{P}^\top \mathbf{S}_2 \mathbf{P} \right\|_{\mathbf{F}}^2 \iff \max_{\mathbf{P}} \mathrm{Tr}(\mathbf{S}_1^\top \mathbf{P}^\top \mathbf{S}_2 \mathbf{P}) = \max_{\mathbf{P}} \langle \mathbf{S}_2^\top \mathbf{P} \mathbf{S}_1, \mathbf{P} \rangle = \max_p p^\top (\mathbf{S}_1 \otimes \mathbf{S}_2)\, p,$$
$$\text{s.t. } \mathbf{P}\mathbf{1} = \mathbf{1}, \mathbf{P}^\top \mathbf{1} = \mathbf{1}, \mathbf{P} \in \{0,1\}^{m \times m}, \tag{7}$$

where $\mathbf{S}_1 \otimes \mathbf{S}_2$ is the Kronecker product of $\mathbf{S}_1$ and $\mathbf{S}_2$, and $\mathbf{S}_1 = \mathbf{Z}_1^\top \mathbf{Z}_1, \mathbf{S}_2 = \mathbf{Z}_2^\top \mathbf{Z}_2$. Therefore, $\mathbf{S}_1$ and $\mathbf{S}_2$ represent the inner graph structures of two anchor sets $\mathbf{A}_1$ and $\mathbf{A}_2$. By minimizing Eq. (7), the inner graph structures can reach a maximum agreement.

**Unified Correspondence Objective:** Taking both first-order feature and second-order structure into consideration, the unified optimization goal for two anchor sets can be formulated as follows,

$$\min_{\mathbf{P}} \|\mathbf{Z}_1 - \mathbf{Z}_2 \mathbf{P}\|_{\mathbf{F}}^2 + \lambda \left\| \mathbf{S}_1 - \mathbf{P}^\top \mathbf{S}_2 \mathbf{P} \right\|_{\mathbf{F}}^2 \iff \max_{\mathbf{P}} \mathrm{Tr}(\mathbf{Z}_1^\top \mathbf{Z}_2 \mathbf{P} + \lambda \mathbf{S}_1^\top \mathbf{P}^\top \mathbf{S}_2 \mathbf{P}),$$
$$\text{s.t. } \mathbf{P}\mathbf{1} = \mathbf{1}, \mathbf{P}^\top \mathbf{1} = \mathbf{1}, \mathbf{P} \in \{0,1\}^{m \times m}, \tag{8}$$

where $\lambda$ is the balanced hypermeter. Problem (8) is a quadratic assignment problem (QAP) and is proved to be NP-hard under most circumstances [12]. The feasible region constraint often relaxes into its convex hull, the Birkhoff polytope with double stochastic region where $\mathbf{P}\mathbf{1} = \mathbf{1}, \mathbf{P}^\top \mathbf{1} = \mathbf{1}, \mathbf{P} \in [0,1]^{m \times m}$. Then the optimization problem transforms into,

$$\max_{\mathbf{P}} \mathrm{Tr}(\mathbf{Z}_1^\top \mathbf{Z}_2 \mathbf{P} + \lambda \mathbf{S}_1^\top \mathbf{P}^\top \mathbf{S}_2 \mathbf{P}), \text{ s.t. } \mathbf{P}\mathbf{1} = \mathbf{1}, \mathbf{P}^\top \mathbf{1} = \mathbf{1}, \mathbf{P} \in [0,1]^{m \times m}, \tag{9}$$

We refer Eq. (9) as the multi-view anchor correspondence framework. To efficiently solve Eq. (9), we adopt the Projected Fixed-Point Algorithm [19] to update $\mathbf{P}$ as follows,

$$\mathbf{P}^{(t+1)} = (1 - \alpha)\mathbf{P}^{(t)} + \alpha \mathbf{\Gamma} \left( \nabla f \left( \mathbf{P}^{(t)} \right) \right) = (1 - \alpha)\mathbf{P}^{(t)} + \alpha \mathbf{\Gamma}(\mathbf{K}^\top + 2\lambda \mathbf{S}_2 \mathbf{P}^{(t)} \mathbf{S}_1^\top), \alpha \in [0,1], \tag{10}$$

where $\alpha$ is the step size parameter, $t$ denotes the number of iterations and $\mathbf{\Gamma}$ denotes the double stochastic projection operator. The convergence of the algorithm has been proved in [19], and we set $\alpha = 0.5$ in this paper. Details of solving Eq. (9) and Remark 2 can be found in supplementary.

**Remark 2.** *The algorithm to solve Eq. (9) converges at rate* $\frac{1}{2} + \lambda \left\| (\mathbf{S}_1 \otimes \mathbf{S}_2) \right\|_F$.

After obtaining $\{\mathbf{P}_i\}_{i=2}^v$, we achieve the fused aligned anchor graph as

$$\mathbf{Z}_{Aligned} = (\mathbf{Z}_1 + \sum_{i=2}^v \mathbf{Z}_i \mathbf{P}_i)/v. \tag{11}$$

The final clustering result can be obtained by simply calculating rank-$k$ truncated SVD on $\mathbf{Z}_{Aligned}$ and output for $k$-means. The whole process is summarized in Algorithm 2.

### 3.3  Complexity Analysis

In this subsection, we analyze our proposed algorithm in terms of space/time complexity.

**Space Complexity:** In our paper, the major memory costs of our method are matrices $\{\mathbf{A}_i\}_{i=1}^{v} \in \mathbb{R}^{m \times d_i}$, $\{\mathbf{P}_i\}_{i=1}^{v} \in \mathbb{R}^{m \times m}$ and $\{\mathbf{Z}_i\}_{i=1}^{v} \in \mathbb{R}^{m \times n}$. Thus the space complexity of our FMVACC is $\mathcal{O}(md + mnv + m^2v)$. In our algorithm, $m \ll n$ and $d \ll n$. Therefore, the space complexity of FMVACC is $\mathcal{O}(n)$.

**Time complexity:** The computational complexity of FMVACC is composed of three steps as mentioned before. When updating $\{\mathbf{Z}_i\}_{i=1}^{v}$, it costs $\mathcal{O}(nmd)$ to get the optimal value. Updating $\{\mathbf{A}_i\}_{i=1}^{v}$ needs $\mathcal{O}(nm^2 + m^2d)$. Then the time cost of alignment module is $\mathcal{O}(m^3)$. Therefore, the total time cost of the optimization process is $\mathcal{O}(nmd + nm^2 + m^2d + m^3)$. Consequently, the computational complexity of our proposed optimization algorithm is linear complexity $\mathcal{O}(n)$.

After the optimization, we perform SVD on $\mathbf{Z}$ to obtain the spectral embedding and output the discrete clustering labels by $k$-means [34]. The post-process needs $\mathcal{O}(nm^2)$, which is also linear complexity respecting to samples. In total, our algorithm achieves MVC with both linear space and time complexity, which demonstrates the efficiency of FMVACC.

### 3.4 Theoretical Analysis

In this subsection, we further show theoretical analysis of our proposed FMVACC with some existing widely-studied multi-view clustering settings.

**Connection with Multi-view Late Fusion Strategy [18]** Multi-view late fusion strategy seeks to fuse partition level information where multiple clustering results can reach a maximum agreement on the final partition matrix. It is easy to show that this strategy is a special case of FMVACC where the $m$ anchors are clustering centers ($m = k$) and each single $\mathbf{Z}_i$ represents the discrete partition. However, the differences are: (i) Late fusion only considers the feature correspondence in Eq. (6) with spectral relaxations $\mathbf{PP}^\top = \mathbf{I}_m$ while the structure counterpart is ignored; (ii) FMVACC shows more generality of late fusion with no limitation of anchor type (clustering centers) and numbers ($k$).

**Connection with PVC [7]** PVC assumes a little different multi-view setting where the data matrices are not aligned with some rows. Taking existing correspondences as supervision signals, PVC recovers other unknown correspondences with a sinkhorn layer in the neural network. We can easily show that PVC is also a special case of FMVACC where only the structure loss in Eq. (7) is measured with some correspondence is known and the anchors are instances themselves.

These two connections with existing late fusion strategy and PVC settings further theoretically demonstrate the generality of FMVACC. We summarize our contributions as: (i) We novelly solve multi-view anchor sets unmeasurable issues by introducing both feature and structure correspondence. (ii) FMVACC is a pioneering work that firstly studies the multi-view anchor unaligned issue for anchor graph fusion tasks. (iii) By following our proposed flexible alignment module, it is quite interesting to rethink current multi-view anchor graph fusion with correct correspondences and further benefit the community.

## 4 Experiments

In this section, we conduct experiments to verify the effectiveness of the proposed FMVACC and alignment module. A simulated dataset is used to verify the AUP issue and the results on seven real-world multi-view datasets are reported. By the way, all the experimental environment are implemented on a desktop computer with an Intel Core i7-7820X CPU and 64GB RAM, MATLAB 2020b (64-bit). More experimental settings can be found in the appendix.

### 4.1 Simulated Multi-view Dataset

**Data Generation** We construct a simulated dual-view dataset, as shown in the left half of Fig. 1. The data have 4 independent clusters where each cluster consists of 50 two-dimensional instances following the respective Gaussian distribution.

**Compared Algorithms** We compare with the two SOTA multi-view anchor graph representatives LMVSC [11] and SFMC [13]. Further, we introduce the proposed flexible anchor selection and alignment modules into the two methods to illustrate the effectiveness of two counterparts.

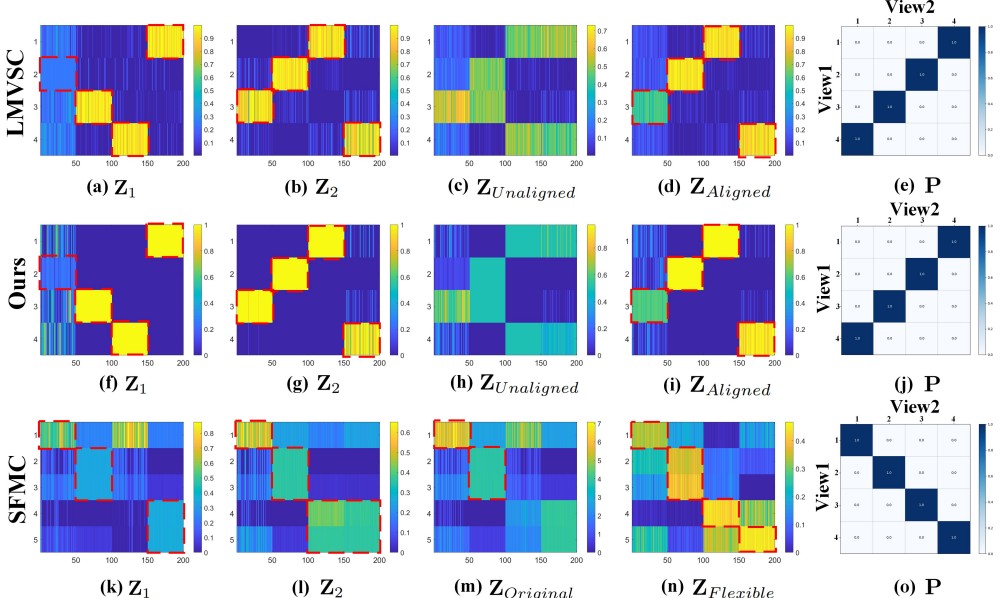

Figure 3: Visualization of the anchor graphs and permutation matrices obtained from LMVSC/SFMC and Ours on the simulated dataset. The first and second rows of LMVSC and Ours clearly verify the effectiveness of the alignment module on better clustering results. The third row illustrates that the flexible strategy with alignment enjoys more preferable performance than the fixed indexes (SFMC).

Table 1: Clustering comparison of LMVSC, SFMC and our FMVACC on simulated dataset (mean$\pm$ std). The better results are highlighted in bold.

| Method | | ACC(%) | NMI(%) | Fscore(%) |
|---|---|---|---|---|
| LMVSC | Unaligned | 84.06±0.16 | 82.18±0.04 | 79.69±0.07 |
| | Aligned | **100.00±0.00** | **100.00±0.00** | **100.00±0.00** |
| Ours | Unaligned | 70.35±1.78 | 72.89±0.68 | 70.28±0.81 |
| | Aligned | **100.00±0.00** | **100.00±0.00** | **100.00±0.00** |
| SFMC | Original | 64.50±0.00 | 80.10±0.00 | 73.41±0.00 |
| | Flexible | **97.50±0.00** | **93.15±0.00** | **95.12±0.00** |

**Visualization and Results Analysis** According to Table 1 and Fig. 3, we have the following observations: (i) For LMVSC and our FMVACC, aligning anchors between views can greatly improve the clustering effect, which verifies the effectiveness of our alignment module. Specifically, LMVSC with our alignment module achieves about 15.94% (ACC), 17.82% (NMI) and 20.31% (Fscore) improvement compared with the column-unaligned results. (ii) From the results of SFMC, it can be seen that flexible anchor selection strategy achieves better clustering performance. Numerically, adopting flexible strategy provides 33.00% (ACC) and 21.71% (Fscore) on the simulated dataset. (iii) For LMVSC and our method in Fig. 3, $\mathbf{Z}_{Aligned}$ obtained after anchors alignment contains less noise and more accurate structural information, and the column permutation matrix $\mathbf{P}$ in the third row is the expected identity matrix $\mathbf{I}$, which verifies the correctness of our method.

## 4.2 Real-world Multi-view Datasets

**Benchmark Datasets** We further perform experiments on eight real-world datasets ranging from 169 to 63896 instances, diverse dimensions and views. 3-Sources contains 169 instances in 6 clusters for 3 views. UCI-Digit refers to 2000 handwritten images in 10 clusters for 3 views. BDGP has 2500 instances for 3 views while SUNRGBD collects 10335 samples. MNIST is a commonly-used dataset and YTF-10/20 are collected from [5].

**Compared Algorithms** Nine state-of-the-art multi-view clustering methods are introduced: MSC-IAS [36], AMGL [21], PMSC [10], RMKM [1], BMVC [46], LMVSC [11], FMCNOF[39], MSGL[9], SFMC [13]. The first three algorithms are graph methods and the others are representative large-scale competitors. Notice that SFMC selects anchor number from $0.1n$ to $n$ so that it can not be applied in large-scale scenarios.

Table 2: Performance on the seven benchmarks. 'OM' indicates the out-of-memory failure.

| Datasets | Samples | MSC-IAS | AMGL | PMSC | RMKM | BMVC | LMVSC | FMCNOF | MSGL | SFMC | Proposed |
|---|---|---|---|---|---|---|---|---|---|---|---|
| ACC (%) | | | | | | | | | | | |
| 3-Sources | 169 | 38.34±3.21 | 20.84±0.25 | 61.34±5.68 | 34.32±0.00 | 47.92±0.00 | 42.36±0.90 | **65.09±0.00** | 46.25±1.18 | 37.27±0.00 | 60.37±7.13 |
| UCI-Digit | 2000 | 82.50±8.08 | 82.26±3.55 | 33.95±1.96 | 81.85±0.00 | 67.00±0.00 | 88.55±3.52 | 54.85±0.00 | 66.78±3.74 | 84.70±0.00 | **89.47±5.41** |
| BDGP | 2500 | 52.09±4.59 | 35.17±1.20 | 26.44±0.19 | 41.44±0.00 | 29.48±0.00 | 50.19±0.03 | 31.08±0.00 | 40.77±0.07 | 20.08±0.00 | **58.63±2.74** |
| SUNRGBD | 10335 | OM | 9.66±0.38 | OM | 18.35±0.00 | 16.94±0.00 | 18.64±0.25 | 19.67±0.00 | 13.74±0.22 | 12.34±0.00 | **22.17±0.66** |
| MNIST | 60000 | OM | OM | OM | 86.21±0.00 | 74.98±0.00 | 95.47±5.39 | 78.29±0.00 | 97.25±2.84 | OM | **98.67±1.93** |
| YTF-10 | 38654 | OM | OM | OM | 75.68±0.00 | 60.43±0.00 | 69.29±4.46 | 43.42±0.00 | 68.30±3.57 | OM | **76.64±4.21** |
| YTF-20 | 63896 | OM | OM | OM | 57.62±0.00 | 60.09±0.00 | 63.02±3.56 | 38.61±0.00 | 65.51±2.36 | OM | **72.54±2.73** |
| NMI (%) | | | | | | | | | | | |
| 3-Sources | 169 | 20.20±2.15 | 8.42±0.22 | 47.53±7.79 | 21.72±0.00 | 46.65±0.00 | 29.79±1.60 | 51.96±0.00 | 26.38±0.42 | 10.92±0.00 | **56.85±5.44** |
| UCI-Digit | 2000 | 87.57±2.85 | 86.14±1.39 | 43.20±1.53 | 76.04±0.00 | 56.69±0.00 | 83.12±1.04 | 57.17±0.00 | 64.89±1.57 | **89.48±0.00** | 84.33±2.37 |
| BDGP | 2500 | 33.07±2.81 | 17.61±1.20 | 3.69±0.19 | 28.12±0.00 | 4.60±0.00 | 25.41±0.07 | 10.29±0.00 | 18.17±0.07 | 2.25±0.00 | **36.81±2.69** |
| SUNRGBD | 10335 | OM | 18.19±0.38 | OM | **26.11±0.00** | 20.39±0.00 | 25.75±0.20 | 15.66±0.00 | 18.76±0.14 | 5.43±0.00 | 19.88±0.61 |
| MNIST | 60000 | OM | OM | OM | 92.08±0.00 | 68.88±0.00 | 95.61±2.02 | 74.49±0.00 | 94.06±1.49 | OM | **96.74±0.75** |
| YTF-10 | 38654 | OM | OM | OM | **80.22±0.00** | 58.91±0.00 | 75.42±1.95 | 39.15±0.00 | 74.71±1.41 | OM | 77.13±1.87 |
| YTF-20 | 63896 | OM | OM | OM | 73.84±0.00 | 71.67±0.00 | 74.02±1.68 | 45.45±0.00 | 74.63±0.82 | OM | **78.47±1.01** |
| Fscore (%) | | | | | | | | | | | |
| 3-Sources | 169 | 29.52±2.05 | 26.91±0.25 | 54.06±5.44 | 30.18±0.00 | 42.77±0.00 | 38.75±1.16 | **59.00±0.00** | 53.02±0.43 | 38.32±0.00 | 54.71±7.67 |
| UCI-Digit | 2000 | 81.67±6.58 | 79.84±3.62 | 28.50±0.75 | 72.10±0.00 | 54.03±0.00 | 80.92±2.28 | 50.40±0.00 | 68.37±2.46 | **83.88±0.00** | 83.33±4.29 |
| BDGP | 2500 | 40.43±2.22 | 34.69±0.77 | 29.55±0.10 | 36.28±0.00 | 26.51±0.00 | 37.82±0.04 | 28.89±0.00 | 43.88±0.05 | 33.15±0.00 | **44.25±2.43** |
| SUNRGBD | 10335 | OM | 6.40±0.20 | OM | 11.68±0.00 | 10.84±0.00 | 11.68±0.14 | **14.08±0.00** | 32.19±0.20 | 12.14±0.00 | 12.94±0.29 |
| MNIST | 60000 | OM | OM | OM | 87.28±0.00 | 62.58±0.00 | 94.84±4.84 | 69.97±0.00 | 97.29±2.70 | OM | **97.65±1.68** |
| YTF-10 | 38654 | OM | OM | OM | 73.27±0.00 | 53.15±0.00 | 64.41±4.05 | 32.88±0.00 | **74.90±1.95** | OM | 71.29±4.22 |
| YTF-20 | 63896 | OM | OM | OM | 53.89±0.00 | 48.06±0.00 | 47.94±3.59 | 25.84±0.00 | **70.70±1.37** | OM | 66.40±2.16 |

**Experimental Results** We conduct comparative experiments of eight multi-view clustering algorithms on seven commonly-used multi-view datasets, and the experimental results are displayed in Table 2. From these results, we observe that:

- Our method also outperforms the other seven algorithms on most clustering metrics when applied to the first three small-scale datasets. Specifically, in terms of ACC, FMVACC achieves 39.53 % (3-Sources), 7.21 % (UCI-Digit) and 23.46 % (BDGP) progress compared to the classical multi-view graph algorithm AMGL.

- When applied on the last four large-scale datasets, compared to well-known large-scale methods RMKM, BMVC, LMVSC, FMCNOF and MSGL, our FMVACC outperforms them on most metrics. Note that the OM failure of SFMC on the last three datasets is due to the large number of anchors it needs to select. The above experimental results show the effectiveness and superiority of our FMVACC.

**Visualization** We visualize the effectiveness of alignment module on UCI-Digit in Fig. 4. For **LMVSC**$_{Unaligned}$ and **LMVSC**$_{Aligned}$ on UCI-digits, the biggest value of similarity values have increased form 0.7 to 0.9, and the Aligned version contains much less noises than unaligned based on correct anchor graph fusion (ACC 66.78% to 82.97%). We can also find similar phenomenon with our proposed approach and conclude that aligned fusion can reduce the noise and enhance clustering performance. For LMVSC and ours, the fused anchor graphs obtained after alignment have clearer structures and therefore achieve better performance, which demonstrates the effectiveness and superiority of our anchor alignment module.

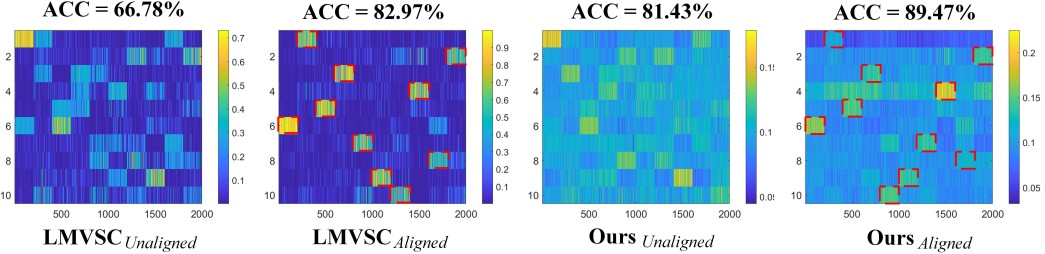

Figure 4: Visualization of the aligned and unaligned anchor graphs (LMVSC and Ours) on UCI-Digit.

**Execution Time:** We evaluate the running time of the proposed FMVACC and seven other MVC comparison algorithms on the benchmark multi-view datasets, which is displayed in Table 3. We can

discover that half of the algorithms cannot be applied on the latter three datasets due to out-of-memory failure. Remarkably, our FMVACC method performs more efficiently than most MVC algorithms, especially on the MNIST dataset with 60000 samples, FMVACC consumes only 571.10 seconds of running time, second only to the BMVC method, while RMKM and LMVSC respectively consume 1468.30 seconds and 604.68 seconds of running times.

Table 3: Running time on the seven benchmarks. 'OM' indicates the out-of-memory failure.

| Datasets | Samples | MSC-IAS | AMGL | PMSC | RMKM | BMVC | LMVSC | FMCNOF | MSGL | SFMC | Proposed |
|----------|---------|---------|------|------|------|------|-------|--------|------|------|----------|
| 3-Sources | 169 | 6.51 | 0.15 | 34.00 | 1.01 | 3.49 | 0.35 | 0.39 | 0.70 | 0.43 | 1.64 |
| UCI-Digit | 2000 | 10.62 | 27.64 | 722.99 | 3.06 | 0.34 | 2.26 | 0.72 | 3.50 | 51.43 | 2.31 |
| BDGP | 2500 | 13.26 | 73.71 | 15215.00 | 7.53 | 0.35 | 2.85 | 1.77 | 3.80 | 38.99 | 5.47 |
| SUNRGBD | 10335 | OM | 4039.10 | OM | 233.92 | 1.79 | 39.94 | 18.23 | 61.40 | 5313.32 | 256.71 |
| MNIST | 60000 | OM | OM | OM | 1468.30 | 17.63 | 604.68 | 63.84 | 684.51 | OM | 571.10 |
| YTF-10 | 38654 | OM | OM | OM | 675.42 | 108.22 | 196.70 | 39.88 | 488.63 | OM | 248.47 |
| YTF-20 | 63896 | OM | OM | OM | 1780.50 | 80.53 | 513.52 | 86.88 | 822.92 | OM | 628.20 |

**Effect of Alignment Module**  In Table 4, we report the accuracy obtained from ablation study on LMVSC and our method. It can be found that for the above two algorithms on all seven benchmarks, the clustering performance achieved by using the alignment module is better than its opponent.

Table 4: Clustering performance comparison of LMVSC and our FMVACC on benchmarks (ACC(%))

| Method | | 3-Sources | UCI-Digit | BDGP | SUNRGBD | MNIST | YTF-10 | YTF-20 |
|--------|--|-----------|-----------|------|---------|-------|--------|--------|
| LMVSC | Unaligned | 46.25±1.18 | 66.78±3.74 | 40.77±0.07 | 13.74±0.22 | 97.25±2.84 | 68.30±3.57 | 65.51±2.36 |
| | Aligned | **47.11±0.97** | **82.97±6.23** | **55.49±0.34** | **17.41±0.39** | **97.85±3.42** | **72.09±2.88** | **67.27±2.84** |
| Ours | Unaligned | 52.81±4.13 | 81.43±5.46 | 49.97±4.17 | 18.00±0.80 | 98.35±1.85 | 71.69±4.80 | 71.45±3.38 |
| | Aligned | **60.37±7.13** | **89.47±5.41** | **58.63±2.74** | **22.17±0.66** | **98.67±1.93** | **76.64±4.21** | **72.54±2.73** |

## 5   Conclusion

In this paper, we present the first study of multi-view anchor unaligned problem. A flexible multi-view anchor graph fusion framework termed **F**ast **M**ulti-**V**iew **A**nchor-**C**orrespondence **C**lustering (FMVACC) is proposed. FMVACC firstly constructs flexible individual anchors and then captures correspondences with both feature and structure information. Theoretical analysis between FMVACC and some existing multi-view strategies is also provided. Extensive experiments are conducted to demonstrate the effectiveness of anchor alignment and our proposed framework FMVACC. It is quite interesting to rethink current multi-view anchor graph fusion with correct correspondences and further benefit the research community. Further, the success of anchor graph clustering heavily depends on the quality of anchor representation. In the future, we will explore how to obtain more-qualified anchors by deep learning.

## Acknowledgment

This work was supported by the National Key R&D Program of China 2020AAA0107100, the Natural Science Foundation of China (project no. 61922088, 61773392, 62006237 and 61976196).

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
