# OpenReview forum: "Align then Fusion: Generalized Large-scale Multi-view Clustering with Anchor Matching Correspondences"
_NeurIPS.cc/2022/Conference — NeurIPS 2022 Accept_

### Official Review · Reviewer_nLLC · 2022-07-08

**Rating:** 6
**Confidence:** 5
**Soundness:** 3 good
**Presentation:** 3 good
**Contribution:** 3 good

**Summary:**

This paper studies an important problem, i.e., anchor-based large-scale multi-view clustering-Anchor Unaligned problem (AUC). The authors review existing multi-view anchor fusion strategies and propose to establish alignment between multi-view anchor sets. The results are promising, and the effectiveness of anchor alignment has been proved.

**Questions:**

1. Although the settings are a little different, the authors could discuss the differences between PVC and the proposed method in details. How can they apply to PVC settings?
2. The paper needs careful proofreading and some typos should be corrected, e.g. Eq.(3). Definitions for the constraints should be presented.
3.The complexity analysis is missing and I suggest the authors should also list with existing baselines (i.e, SFMC and LMVSC) both space and time complexity.
4. Figure (e)(j)(o) in Figure 3 are the obtained permutation matrices and more discussions can be introduced.
5. The authors should survey not only multi-graph clustering models but also with other multi-view approaches in the part of related work.


**Limitations:**

YES

**Strengths And Weaknesses:**

Strength:
1.	The idea is interesting and novel. Figure 1 and Table 1 illustrate the issue of anchor-unaligned problem for existing multi-view anchor graph clustering.
2.	This paper introduces a flexible anchor graph fusion framework termed FMVACC to tackle the AUP problem, which is a generalized large-scale multi-view anchor graph study.
3.	The authors solve the unmeasurable multi-dimensional anchor matching problem by introducing two parts: feature and structure correspondences in Algorithm 2.
4.	The experimental results seem promising. The effectiveness of the proposed anchor-aligned module has been proved in Table 1/3 and Figure 3/4.
5.	The code is available and easy to reproduce.

Weakness:
1.	The complexity analysis is missing and I suggest the authors should also list with existing baselines (i.e, SFMC and LMVSC) both space and time complexity.
2.	The paper needs careful proofreading and some typos should be corrected.
3.	For better representation, Figure (e)(j)(o) in Figure 3 are the obtained permutation matrices and more discussions can be introduced.

---

> ### Author Response · Authors · 2022-08-01
> **Response to reviewer nLLC**
>
> Thank you very much for your helpful suggestions and interest in our learning problem.We have taken the  constructive comments and suggestions into consideration and the detailed responses to the comments and the corresponding revisions are summarized below.
>
> Q1:. The complexity analysis is missing and I suggest the authors should also list with existing baselines (i.e, SFMC and LMVSC) both space and time complexity.
>
> A1:Thanks for your advice. The detailed complexity analysis is provided at section A.4 in Appendix.
>
> **Space Complexity:** The space complexity of our FMVACC is $\mathcal{O}(md+ mnv+ m^2v)$. In our algorithm, $m \ll n$ and $d \ll n$. Therefore, the space complexity of FMVACC is $\mathcal{O}(n)$.
>
> **Time complexity:** The total computational complexity of FMVACC is $\mathcal{O}(nmd+nm^2+m^2d+m^3)$, which is linear complexity $\mathcal{O}(n)$.
>
> After the optimization, we perform SVD on $\mathbf{Z}$ to obtain the spectral embedding and output the discrete clustering labels by $k$-means. The post-process needs $\mathcal{O}(nm^2)$, which is also linear complexity respecting to samples. In total, our algorithm achieves MVC with both linear space and time complexity, which demonstrates the efficiency of FMVACC.
>
> |  Methods | Space Complexity | Time Complexity |
> |:--------:|:----------------:|:---------------:|
> |  MSC-IAS |     $\mathcal{O}(vn^2)$             |    $\mathcal{O}(n^3)$             |
> |   AMGL   |     $ \mathcal{O}(v n^2 + nk)$             |        $\mathcal{O}(n^3)$         |
> |   PMSC   |     $ \mathcal{O}(vn^2+vnk)$             |     $\mathcal{O}(n^3)$            |
> |   RMKM   |     $ \mathcal{O}((n+d)m)$             |       $\mathcal{O}(ndk)$          |
> |   BMVC   |      $\mathcal{O}(nm+nk)$            |             $\mathcal{O}(nmv+nkm+nk)$ |
> |   LMVSC  |      $ \mathcal{O}(vk(n + d))$            |      $\mathcal{O}(nm^3)$            |
> |   SFMC   |      $\mathcal{O}(nmv)$            |    $\mathcal{O}(nm^3+v^3)$               |
> | Proposed |     $\mathcal{O}(md+ mnv+ m^2v)$             |      $\mathcal{O}(nmd+(n+d)m^2+m^3)$           |
>
>
> where $n$,$v$,$m$ are the numbers of samples views and anchors among each view.$d = \sum_{i}^{v} d_i$ is the sum of view dimensions.
>
> Q2:For better representation, Figure (e)(j)(o) in Figure 3 are the obtained permutation matrices and more discussions can be introduced.
>
> A2: Thanks for the comments. In Fig.3 , the three figures (e)(j) and (o) are the obtained permutation matrices by solving Eq. (9) for LMVSC, Our algorithm and SFMC respectively.  The two figures (e) and (j) clearly show the obtained correspondences between the two views for ours and LMVSC. The anchors order (1234) in view 1 match with anchor order(4321) where cna be verified in the obtained anchor graphs in the two views. Moreover, since SFMC adopts the samples with same indexes as anchors, the idea permutation matrix is identity matrix $\mathbf{I}_m$ where also verified by figure (o). The results clearly illustrate the effectiveness of our proposed alignment module.
>
> Q3:Although the settings are a little different, the authors could discuss the differences between PVC and the proposed method in details. How can they apply to PVC settings?
>
> A3: We summarize the difference between our method and PVC as follows:
>
> 1.**Problem is different:** PVC is proposed to handle with **unmapped** multi-view data where some correspondences have been given as supervision signals. Our paper aims at solving the large-scale multi-view anchor-unaligned issue.
>
>
> 2.**Setting is different:** PVC proposes to capture sample mapping with the supervision of some existing known correspondence. Hence it is regarded as a **semi-supervised(some mappings are known)**. However, our paper is designed to solve the anchor-unaligned issue in **multi-view multi-view graph clustering with anchors**.
>
> 3.**Method is different:** PVC firstly projects multi-dimensional data into the latent common space by auto-encoder network. However, ours is totally unsupervised and seeks correspondences among each original view. Ours combines feature and structure correspondence to fuse multi-view anchor graphs.
>
>
> **How to extend Our FMVACC to PVC problem:**  We can easily show that PVC is also a special case of our proposed FMVACC. Taking the $n$ samples as anchors in each view, the single-view anchor graph extends to the full graph with size of $n \times n$. Then the unmapped data in PVC can be solved with the structural loss in our formulation where some entries of the permutation matrix $\mathbf{P}$ have been set as 1 by the pre-known correspondences.

---

> > ### Author Response · Authors · 2022-08-01
> > **Further response to Reviewer nLLC**
> >
> > Q4:The authors should survey not only multi-graph clustering models but also with other multi-view approaches in the part of related work.
> >
> > A4:Thanks for your constructive suggestions. In the introduction and related work sections, we enlarge our literature review with more strategies:multi-view NMF and multi-view ensemble clustering. Multi-view NMF adopts matrix factorization to learn the shared spaces with designed properties (i.e. sparse and low-rank) [1][2]. Multi-view ensemble clustering aims to fuse multiple partition results into a more robust and impressive clustering results [3,4]. We will update our 'Related work' section with the two afore-mentioned styles of multi-view clustering.
> >
> > [1]Zhao H, Ding Z, Fu Y. Multi-view clustering via deep matrix factorization[C]//Thirty-first AAAI conference on artificial intelligence. 2017.
> >
> > [2]Brbić M, Kopriva I. Multi-view low-rank sparse subspace clustering[J]. Pattern Recognition, 2018, 73: 247-258.
> >
> > [3]Tao Z, Liu H, Li S, et al. From ensemble clustering to multi-view clustering[C]//IJCAI. 2017.
> >
> > [4]Liu H, Wu J, Liu T, et al. Spectral ensemble clustering via weighted k-means: Theoretical and practical evidence[J]. IEEE transactions on knowledge and data engineering, 2017, 29(5): 1129-1143.
> >
> > Q5:The paper needs careful proofreading and some typos should be corrected, e.g. Eq.(3). Definitions for the constraints should be presented.
> >
> > A5:Thanks for the constructive comments. We have carefully revised our paper to make the new version more readable. Please check our revised manuscript.

---

> > > ### Comment · Reviewer_nLLC · 2022-08-08
> > > **thank for the reply**
> > >
> > > Thanks for the detailed reply and new results, I satisfy with the response and would keep my recommendation.

---

### Official Review · Reviewer_U2ZL · 2022-07-09

**Rating:** 7
**Confidence:** 4
**Soundness:** 4 excellent
**Presentation:** 3 good
**Contribution:** 3 good

**Summary:**

This paper focuses on the large-scale multi-view anchor graph fusion strategies where the anchor graphs in individual views are not naturally aligned. The authors discover this phenomenon and provide a matching framework to address this issue. The paper is well-written and easy to read. The idea is well-motivated with clear contributions of illustrating necessity of multi-view anchor alignment. The comprehensive experimental results not only show performance improvements with the proposed FMVACC but also enhances existing large-scale baselines with more flexibility, indicating the effectiveness of multi-view anchor alignment.

**Questions:**

1.Some details should be clarified in Table 2 with representative large-scale multi-view methods. RMKM is not a graph method and should be introduced in the related work sections.
2.The authors should illustrate how to initialize single-view anchors since initializations are important.
3.There are some formatting inconsistencies in the text that the authors should check carefully, i.e. AUP and \mathcal for complexity analysis.
4.Some other strategies of multi-view clustering should be discussed in the introduction part, for example, multi-view NMF and multi-view ensemble clustering.
5.I suggest the authors can compare the performance of some latest multi-view scalable graph clustering methods [1][2]. [1] is an extension version of LMVSC mentioned in the manuscript and [2] utilizes NMF into multi-view anchor graph study.
[1]Structured Graph Learning for Scalable Subspace Clustering: From Single View to Multiview. IEEE TCYB, 2021.
[2]Fast Multi-View Clustering via Nonnegative and Orthogonal Factorization. IEEE TIP, 2021.


**Limitations:**

Yes. The authors have adequately addressed the limitations and potential negative societal impact of their work.

**Strengths And Weaknesses:**

Strength:
1.	The paper is well-written and easy to read. The idea of recognizing the multi-view anchor alignment problem has been overlooked in the existing literature which it can benefit further research and community.
2.	The authors propose the first study of generalized flexible multi-view anchor graph framework. Different from sampling fixed anchors for AUP problem in SOTA, a matching framework has been made with more flexibility and performance improvement.
3.	The effectiveness of the alignment module has been proved both on simulated and real-world datasets. Moreover, existing baselines enjoy considerable performance improvement with the module.
4.	The method can be applied to large-scale scenarios where the scalability is proved by theory analysis and experiments.
5.	Experiments compared with recent works and ablation studies are well presented.

Weakness:
1.	Although the proposed method focuses on multi-view graph method, the literature review can be enlarged with other strategies of multi-view clustering approaches, for example, multi-view NMF and multi-view ensemble clustering.
2.	The authors should illustrate how to initialize single-view anchors since initializations are important.
3.	There are some formatting inconsistencies in the text that the authors should check them carefully, i.e. AUP.

---

> ### Author Response · Authors · 2022-08-01
> **Response to Reviewer U2ZL**
>
> We thank Reviewer U2ZL for careful reading and constructive suggestions. We have taken the constructive comments and suggestions into consideration and the detailed responses to the comments and the corresponding revisions are summarized below.
>
> Q1: Although the proposed method focuses on multi-view graph method, the literature review can be enlarged with other strategies of multi-view clustering approaches, for example, multi-view NMF and multi-view ensemble clustering. Some details should be clarified in Table 2 with representative large-scale multi-view methods. RMKM is not a graph method and should be introduced in the related work sections.
>
> A1: Thanks for your suggestions. In the introduction and related work sections, we enlarge our literature review with more strategies:multi-view NMF and multi-view ensemble clustering. Multi-view NMF adopts matrix factorization to learn the shared spaces with designed properties (i.e. sparse and low-rank)[1][2]. Multi-view ensemble clustering aims to fuse multiple partition results into a more robust and impressive clustering result[3,4]. We will update our 'Related work' section with the two afore-mentioned styles of multi-view clustering.The chosen algorithms in Table 2 (RMKM, BMVC, LMVSC and SFMC) are designed for large-scale multi-view clustering tasks. RMKM [5] adaptively learns the clustering indicator matrix via following k-means formulation and avoid the huge computational burden.
>
> [1]Zhao H, Ding Z, Fu Y. Multi-view clustering via deep matrix factorization[C]//Thirty-first AAAI conference on artificial intelligence. 2017.
>
> [2]Brbić M, Kopriva I. Multi-view low-rank sparse subspace clustering[J]. Pattern Recognition, 2018, 73: 247-258.
>
> [3]Tao Z, Liu H, Li S, et al. From ensemble clustering to multi-view clustering[C]//IJCAI. 2017.
>
> [4]Liu H, Wu J, Liu T, et al. Spectral ensemble clustering via weighted k-means: Theoretical and practical evidence[J]. IEEE transactions on knowledge and data engineering, 2017, 29(5): 1129-1143.
>
> [5]Cai X, Nie F, Huang H. Multi-view k-means clustering on big data[C]//Twenty-Third International Joint conference on artificial intelligence. 2013.
>
> Q2: The authors should illustrate how to initialize single-view anchors since initializations are important.
>
> A2: Different with SFMC [6] sampling multi-view samples with same indexes as anchors to avoid AUP issue,  we propose to generate **flexible** anchors among each view. The anchors are firstly initialized with $k$-means centroids and then refined by the optimization algorithm 1.
>
> [6]Li X, Zhang H, Wang R, et al. Multiview clustering: A scalable and parameter-free bipartite graph fusion method[J]. IEEE Transactions on Pattern Analysis and Machine Intelligence, 2022, 44(1): 330-344.
>
> Q3: There are some formatting inconsistencies in the text that the authors should check them carefully, i.e. AUP.
>
> A3: Thanks for your careful reading. We will correct the typos in the revised version.
>
>
> Q4: I suggest the authors can compare the performance of some latest multi-view scalable graph clustering methods [7][8]. [7] is an extension version of LMVSC mentioned in the manuscript and [8] utilizes NMF into multi-view anchor graph study.
>
> A4:Thanks for your constructive comments. We have reviewed the mentioned two papers[7,8]. [7] generate anchors in each view and then obtain a unified low-rank anchor graph. It is noticeable that [7] still suffer form AUP issue since anchors are independeent among each views. [8] constructs individual anchor graphs and then adopts the matrix factorization to get the clustering labels. We also conduct comparison experiments and show the results as follows,
>
> | ACC         | UCI-Digit | BDGP      | SUNRGBD   | MNIST     | YTF-10    | YTF-20    |
> |-------------|-----------|-----------|-----------|-----------|-----------|-----------|
> | FMCNOF      | 54.85     | 31.08     | 19.67     | 78.29     | 43.42     | 38.61     |
> | MSGL        | 72.18     | 53.17     | 19.61     | **97.68** | 70.68     | 63.54     |
> | Proposed    | **83.59** | **59.51** | **21.95** | 94.39     | **74.80** | **71.32** |
> | NMI（mean） | UCI-Digit | BDGP      | SUNRGBD   | MNIST     | YTF-10    | YTF-20    |
> | FMCNOF      | 57.17     | 10.29     | 15.66     | 74.49     | 39.15     | 45.45     |
> | MSGL        | 74.83     | 26.98     | **26.21** | 93.72     | 70.68     | 63.54     |
> | Proposed    | **81.13** | **35.55** | 19.22     | **95.11** | **76.53** | **77.89** |
>
> From the table, our proposed method also enjoys more preferable task performance on the benchmark datasets. It is expected we can update our anchor generation strategy to further improve the clustering performance in future work.
>
> [7]Structured Graph Learning for Scalable Subspace Clustering: From Single View to Multiview. IEEE TCYB, 2021.
>
> [8]Fast Multi-View Clustering via Nonnegative and Orthogonal Factorization. IEEE TIP, 2021.

---

> > ### Comment · Reviewer_U2ZL · 2022-08-10
> > **resopnd to rebuttal**
> >
> > Thank you for your response. I agree that the idea of recognizing the multi-view anchor alignment problem has been overlooked in the existing literature which can benefit further large-scale multi-view clustering research.
> >
> > I think the authors have addressed all of my concerns. I will keep my score and suggest acceptance.

---

### Official Review · Reviewer_Rr2L · 2022-07-11

**Rating:** 6
**Confidence:** 5
**Soundness:** 3 good
**Presentation:** 2 fair
**Contribution:** 3 good

**Summary:**

This paper focuses on multi-view graph clustering. The author studied a practical problem in multi-view graph clustering, i.e., anchor-unaligned problem. To solve this, the author propose a new anchor graph fusion framework including an anchor alignment module to solve AUP. Experiments on several datasets show reasonable performance improvement and effectiveness of the proposed method.

**Questions:**

1. What is the difference between Eq.8 and Eq.9？Detailed proof is expected.
2. In training, the permutation matrix is computed by solving Eq.9, which is actually relaxed. How to get the permutation matrix (binary value) in the inference period?

**Strengths And Weaknesses:**

Strengths:
1. The motivation is strong and practical.
2. The solution to obtain the permutation matrix is interesting, which is proved effective theoretically and experimentally.
3. This paper is the first work to study the AUP in multi-view graph clustering.

Weaknesses:
1. The reviewer suggests improving Fig.2 for better clarity. Current figures cannot deliver the main idea of the proposed method.
2. The authors do not clearly define the experimental settings in the real-world multi-view datasets and how they tune the parameters. Do these datasets still suffer from the AUP problem?
3. The visualizations of the anchor graphs on UCI digits show little difference between the aligned and unaligned cases, which is less convincing.

---

> ### Author Response · Authors · 2022-07-31
> **Response to Reviewer Rr2L**
>
> We thank Reviewer Rr2L for identifying the contribution of AUP problems in large-scale MVC. We have taken the  constructive comments and suggestions into consideration and the detailed responses to the comments and the corresponding revisions are summarized below.
>
> Q1:The authors do not clearly define the experimental settings in the real-world multi-view datasets and how they tune the parameters. Do these datasets still suffer from the AUP problem?
>
> A1: Thanks for the comments. The **Anchor-Unaligned Problem (AUP) naturally** exists in large-scale multi-view datasets. To improve the efficiency of multi-view graph clustering methods, some anchors/landmarks are selected to replace the $n \times n$ large graph into small anchor graphs $ n \times m$. However, the anchors sets across multiple views are unaligned (Shown in Fig. 3 and 4) and therefore leads to incorrect anchor graph fusion.
>
> Moreover, we emphasize the difference between ours and PVC[1]/MVC-UM[2] as follows:
>
> 1.**Problem is different:** PVC and MVC-UM are proposed to handle with **unmapped** multi-view data where the multi-view data are disordered. However, this paper aims at solving the large-scale multi-view anchor-unaligned issue. When existing fast large-scale multi-view clustering methods generate anchor independently among each view, the  **Anchor-Unaligned Problem (AUP) naturally** exists.
>
>
> 2.**Setting is different:** PVC[1] proposes to capture sample mapping with the supervision of some existing known correspondences. Hence it is regarded as a **semi-supervised(some mappings are known)**. However, our paper is designed to solve the **unsupervised large-scale multi-view clustering with anchors**.
>
> 3.**Method is different:** [1] and [2] firstly projects multi-dimensional data into the latent common space by neural networks or matrix factorization. However, ours is totally unsupervised and seeks correspondences among each original view. Our proposed method combines both feature and structure correspondence to fuse multi-view anchor graphs. Extensive experiments on  benchmark datasets demonstrate the effectiveness and our proposed multi-view anchor correspondence framework. Moreover, the proposed anchoralignment module also shows significant performance improvement applying to existing multi-view anchor graph competitors indicating the importance of anchor alignment.
>
>
> [1]Huang Z, Hu P, Zhou J T, et al. Partially view-aligned clustering[J]. NIPS2020, 33: 2892-2902.
>
> [2]Yu H, Tang J, Wang G, et al. A Novel Multi-View Clustering Method for Unknown Mapping Relationships Between Cross-View Samples[C]//KDD2021: 2075-2083.
>
> Q2:The visualizations of the anchor graphs on UCI digits show little difference between the aligned and unaligned cases.
>
> A2: Thanks for your careful reading. In $LMVSC_{Unaligned}$ and $LMVSC_{aligned}$ on UCI digits, you can find that the biggest value of similarity values have increased form 0.7 to 0.9, and the Aligned version contains much less noises than unaligned based on correct anchor graph fusion (ACC 63.37% to 80.87%). We can also find similar phenomenon with our proposed approach and conclude that **aligned** fusion can reduce the noise and enhance clustering performance.
>
> Q3:The reviewer suggests improving Fig.2 for better clarity. Current figures cannot deliver the main idea of the proposed method.
>
> A3: Thanks for the suggestion. We will rearrange the lines and give optimization goals on the right of Figure 2 to better illustrate our idea.
>
> Q4:What is the difference between Eq.8 and Eq.9？Detailed proof is expected.
>
> A4: Thanks for the comment. The difference between Eq. (8) and Eq. (9) is the constraint on the matrix $\mathbf{P} \in \mathbb{R}^{m \times m}$. In Eq. (8), $\mathbf{P}$ is the permutation matrix where $\mathbf{P} \in \{0,1\}^{m \times m}$. Then, we relax the constraint  into its convex hull, the Birkhoff polytope with double stochastic region $\mathbf{P} \mathbf{1} =\mathbf{1},  \mathbf{P}^{\top}\mathbf{1}=\mathbf{1}, \mathbf{P} \in [0,1]^{m \times m}$. Please check our revision in the 196-th line of the manuscipt.
>
> Q5:In training, the permutation matrix is computed by solving Eq.9, which is actually relaxed. How to get the permutation matrix (binary value) in the inference period?
>
> A5:After obtaining the matrix $\mathbf{P}$ by solving Eq. (9), we apply the Sinkhorn operator to $\mathbf{P}$ and get the binary permutation matrix [3]. We will add the illustration in the revision version.
>
> [3]Mena G, Belanger D, Linderman S, et al. Learning Latent Permutations with Gumbel-Sinkhorn Networks[C]//International Conference on Learning Representations. 2018.

---

### Author Response · Authors · 2022-08-09
**Paper Discussion**

Dear Reviewers, Area Chairs, Senior Area Chairs and Program Chairs,

We sincerely thank the efforts and constructive comments you have made for this paper. The reviewers put forward many insightful questions and valuable suggestions towards improving our paper.

In the rebuttal phase, we provided detailed responses to all reviewers' comments point by point, hoping to address the issues raised by reviewers Rr2L, U2ZL and nLLC, including more detailed comparison of existing partially-aligned multi-view setting (PVC), illustration of experimental results and added experiments of mentioned SOTA algorithms. Moreover, we indicate that the **Anchor-Unaligned Problem (AUP)** naturally exists in large-scale multi-view clsutering with view-independent anchors.

The discussion period is coming to an end, and we are actively awaiting for further discussion from Reviewers. If you have any questions, we are happy to discuss them with you at any time.

Thanks & Regards,

Authors of paper-228.

---

### Meta-Review · Area_Chair_2PzL · 2022-08-25

**Recommendation:** Accept
**Confidence:** Certain

**Metareview:**

All reviewer agree that this paper is innovative and well-written, so I recommend to accept.

**Award:**

No

---

### Decision · Program_Chairs · 2022-09-14

Accept